# Fecal Microbiota Transplantation Derived from Alzheimer’s Disease Mice Worsens Brain Trauma Outcomes in Wild-Type Controls

**DOI:** 10.3390/ijms23094476

**Published:** 2022-04-19

**Authors:** Sirena Soriano, Kristen Curry, Qi Wang, Elsbeth Chow, Todd J. Treangen, Sonia Villapol

**Affiliations:** 1Department of Neurosurgery and Center for Neuroregeneration, Houston Methodist Research Institute, 6670 Bertner Avenue, Houston, TX 77030, USA; ssoriano@houstonmethodist.org (S.S.); elsbethchow@gmail.com (E.C.); 2Department of Computer Science, Rice University, Houston, TX 77005, USA; kdc10@rice.edu (K.C.); qiwangrice@gmail.com (Q.W.); treangen@rice.edu (T.J.T.); 3Department of Neuroscience in Neurological Surgery, Weill Cornell Medical College, New York, NY 10065, USA

**Keywords:** microbiome, traumatic brain injury, Alzheimer’s disease, fecal microbiota transplant, neuroinflammation, microglia, astrocytes, dysbiosis, *Muribaculum*, *Lactobacillus johnsonii*

## Abstract

Traumatic brain injury (TBI) causes neuroinflammation and neurodegeneration, both of which increase the risk and accelerate the progression of Alzheimer’s disease (AD). The gut microbiome is an essential modulator of the immune system, impacting the brain. AD has been related with reduced diversity and alterations in the community composition of the gut microbiota. This study aimed to determine whether the gut microbiota from AD mice exacerbates neurological deficits after TBI in control mice. We prepared fecal microbiota transplants from 18 to 24 month old 3×Tg-AD (FMT-AD) and from healthy control (FMT-young) mice. FMTs were administered orally to young control C57BL/6 (wild-type, WT) mice after they underwent controlled cortical impact (CCI) injury, as a model of TBI. Then, we characterized the microbiota composition of the fecal samples by full-length 16S rRNA gene sequencing analysis. We collected the blood, brain, and gut tissues for protein and immunohistochemical analysis. Our results showed that FMT-AD administration stimulates a higher relative abundance of the genus *Muribaculum* and a decrease in *Lactobacillus johnsonii* compared to FMT-young in WT mice. Furthermore, WT mice exhibited larger lesion, increased activated microglia/macrophages, and reduced motor recovery after FMT-AD compared to FMT-young one day after TBI. In summary, we observed gut microbiota from AD mice to have a detrimental effect and aggravate the neuroinflammatory response and neurological outcomes after TBI in young WT mice.

## 1. Introduction

Alzheimer’s disease (AD) is characterized by progressive cognitive and motor impairment associated with the accumulation of beta-amyloid (Aβ) protein and tau protein deposition [1]. Before any cognitive impairment is observed, an initial cellular pathology that includes alterations in neurons, microglia, astroglia, and vasculature develops in parallel to the amyloid β accumulation [2]. Many processes, such as dysregulated protein clearance and lipid metabolism, mitochondria and peroxisome dysfunction, oxidative damage, and neuroinflammation processes, have proven causative of the cellular pathology that ultimately results in cell death [3,4,5,6]. The clinical signs of AD follow the cellular phase of the disease, progressing from mild cognitive impairment to severe dementia and detrimental behavioral changes [7]. AD biomarkers in cerebrospinal fluid (CSF), blood, and neuroimaging techniques are being developed to achieve early diagnosis in clinical practice [8,9]. However, current therapeutic developments for AD have shown modest clinical benefits so far [10].

Traumatic brain injury (TBI) is a well-established risk factor for AD [11]. Specifically, numerous studies demonstrate that the risk of developing AD is higher in patients who have previously suffered brain trauma [12,13,14,15]. TBI has a devastating effect on the brain, causing both short- and long-term damage that is not effectively alleviated by traditional pharmacological treatments [16,17]. The initial insult is followed by a complex pathophysiological progression that involves distinct molecular and cellular pathways, triggering the neuroinflammatory response [18]. Although there is a strong epidemiological correlation between TBI and the later development of AD, the mechanisms driving this link have yet to be fully elucidated. The increased neuroinflammation and neuronal loss present in AD brains can be exacerbated by TBI [19]. Further, preclinical studies show that amyloid precursor protein (APP), Aβ, and pathological tau are increased after TBI in mice [20,21]. Aβ plaques have also been found in patients within hours following TBI [22,23]. 

Not only does TBI increase the risk of developing dementia, but the progression of AD is more rapid in those who have previously suffered from brain trauma [24,25,26]. In older adults, TBI often results in an accumulation of neuroinflammation that can exacerbate the neuropathological changes and contribute to the acceleration of dementia-related pathologies, including AD [27,28]. This is of particular relevance since TBI displays a peak incidence in the older adult population, which is largely affected by AD [29]. The cognitive and motor impairment characteristic of AD further contributes to the already high occurrence of TBI in the elderly. Moreover, older individuals have extended recovery periods compared to their younger peers because of different neuroinflammatory responses [30]. The increased risk and worsened prognosis for recovery from brain injury indicates a need for treatments targeted to senior TBI patients, with special attention in the context of AD-type dementia.

The inflammatory responses that are triggered by both AD and TBI extend past the brain and into the periphery, and they potentially include inflammation from the immune system in the gut [31]. The gut–brain axis (GBA) mediates the bidirectional communication between the gastrointestinal tract and the brain through neural, immune, endocrine, and metabolic signaling pathways [32]. The gut microbiome not only is a key component of the GBA, but it also plays an essential role in immunity. A decrease in the diversity of gut microbiota has been shown to significantly increase toxic components in the bloodstream [33,34] that may reach the brain and lead to neuroinflammation. Microbiota dysbiosis also causes a decrease in bacterial metabolites, such as gamma-aminobutyric acid (GABA) and short-chain fatty acids (SCFAs), which are considered key mediators for microbiota–microglia interaction [35]. These compounds have the potential to translocate from the mucosa to systemic circulation and affect brain function.

Changes in the gut microbiota composition have been associated with aging and many neurological and psychiatric disorders [36,37,38,39]. Decreased microbiota diversity and dysbiosis in the gut of aged individuals causes the release of lipopolysaccharides (LPS) and inflammatory cytokines that disrupt the blood–brain barrier and promote neuroinflammation and neuronal injury, leading to accelerated neurodegeneration [40]. Brain injury also decreases bacterial diversity in young and old patient populations; moreover, a less diverse gut microbiome has been shown to increase both susceptibility to brain damage and recovery length [41]. Similarly, clinical studies have reported a lack of diversity in the fecal microbiota of AD patients together with alterations in the gut microbiota composition [42]. Specifically, the microbiota of aged individuals with AD have lower levels of butyrate-producing bacteria [43], which could lead to increased neuroinflammation and cognitive loss. Moreover, an increase in gram-negative species such as Bacteroidetes has been found to correlate with CSF amyloid and tau [44]. LPS and bacterial amyloids synthesized by the gut microbiota can trigger immune cells by binding microglia through toll-like receptors (TLR), leading to an M1 phenotype and pro-inflammatory state, and thus resulting in neuroinflammation and neuronal death. LPS is three times higher in AD patients’ plasma compared to healthy controls [45]. Additionally, AD can weaken the gastrointestinal barrier and promote a proinflammatory phenotype that involves intestinal microbes [46]. Colonic inflammation and intestinal barrier disruption have been found in about 70% of AD patients, as suggested by elevated fecal calprotectin levels [47]. Similarly, the production of amyloid by bacteria or fungi in the gut microbiota may induce neurodegeneration and inflammation [48,49]. 

Although gut dysbiosis has been linked with both TBI and AD, little is known about its role on AD patients suffering from TBI. Therefore, in this study, our goal was to determine whether AD microbiota would negatively impact on TBI outcomes. Our results show that delivering fecal microbiota transplantation (FMT) from AD mice into wild-type (WT) mice after TBI exerts a detrimental effect on the neurodegenerative and neuroinflammatory processes and motor recovery. To the best of our knowledge, this is the first study to describe how the gut microbiome from aged AD mice worsens inflammatory results after TBI. Our study may help elucidate why neurodegenerative processes after brain injury are more severe in AD patients. Further, they illustrate the need for targeted therapeutic strategies aimed at restoring the bacterial flora of AD patients to reduce the inflammatory response that can be accelerated by brain damage.

## 2. Results

### 2.1. FMT-AD Induced Gut Microbiome Changes in C57BL/6 Recipient Mice Following TBI

To assess the effect of AD microbiota on recovery following TBI, we transplanted gut microbiota-derived from 3×Tg-AD mice into young C57BL/6 (WT) mice after brain injury. We prepared FMT from aged 3×Tg-AD (FMT-AD) and young WT (FMT-young) mice. Young WT male and female animals received their designated FMT-young or FMT-AD by oral gavage after controlled cortical impact (CCI) injury (Figure 1). 

At 3 days post-injury, microbial DNA was extracted from cecum samples of the recipient mice, and next generation 16S rRNA gene sequencing was performed using the Nanopore MinION platform. We investigated the changes in the gut microbiome composition of the mice that received FMT-young compared to FMT-AD at the phylum and family levels by representing their relative abundances in stacked bar plots (Figure 2a,b). Some shifts were observed at the phylum level on the FMT-AD mice, mainly a reduction in Firmicutes together with an increase in Bacteroidetes (Figure 2a). However, the Firmicutes/Bacteroidetes ratios of the FMT-young and FMT-AD groups did not significantly differ (Mann–Whitney test *p* = 0.139, mean (±SEM) FMT-young = 118.69 (±39.40), FMT-AD = 45.17 (±12.02)). Further changes were evident at the family level in the FMT-AD recipients compared to FMT-young, including a decrease in *Lachnospiraceae* and *Clostridiaceae* paired with a relative increase in *Eubacteriaceae*, *Oscillospiraceae*, and *Muribaculaceae* (Figure 2b). At the genus level, *Muribaculum* (*q* = 0.02) significantly increased relative abundance in FMT-AD mice, and the species *Lactobacillus johnsonii* (*q* = 0.13) reduced relative abundance, albeit not significantly (Figure 2c). However, there was no significant difference between the groups (Figure 2d). Further, the alpha diversity at the species level based on the Simpson and Shannon metrics did not significantly differ between the time points (*p* = 0.178 and *p* = 0.425, respectively) (Figure 2e). Collectively, these results provide evidence that the fecal microbiota of FMT-AD is different from that of FMT-young, which led us to the hypothesis that this difference may affect the neuroinflammatory responses following TBI.

### 2.2. Microbiota from AD Mice Aggravated the Lesion Size after TBI

To evaluate whether the gut microbiota from AD mice has any effect during the acute phase after TBI in WT mice, we assessed the cortical lesion size using cresyl-violet stained brain sections. We found a significant increase in mean lesion volume (*p* = 0.022) when calculated as a percent of total ipsilateral hemisphere volume in FMT-AD mice compared to FMT-young mice at 3 days post-TBI. This was confirmed by a significant increase in lesion score (*p* = 0.002) on a scale from 0 to 4 in FMT-AD mice compared to FMT-young mice (Figure 3a–c).

### 2.3. Microbiota from AD Mice Impaired Motor Ability after TBI

Motor dysfunction is a major consequence of TBI, and several motor assessments have been performed using the CCI injury model [50,51]. To investigate whether fecal transplant affects motor outcomes, we employed a rotarod test to compare motor skills in mice that received FMT from AD mice and those that received FMT from young animals. Our results indicate that, regardless of sex, motor function was significantly impaired in the FMT-AD group compared to the FMT-young group at 1 day after injury (*p* = 0.0037) (Figure 3d).

### 2.4. FMT-AD Led to Increased Neuroinflammation

Glia activation in the injured brain is detrimental at 3 days post-TBI [52]. Therefore, we analyzed changes in the activation of microglia, astrocytes, and neutrophil infiltration by detecting Iba-1 (microglia/macrophages), GFAP (astrocytes), and Ly6B.2 (neutrophil) positive cells, respectively (Figure 4). Our results show an overall higher inflammatory response after FMT-AD compared to FMT-young. Specifically, there was increased microglia and macrophage activation and astroglia activation, but no change in neutrophil infiltration into the pericontusional cortex. Quantification of activated cells showed a significantly higher percent area of tissue occupied by Iba1- and GFAP-positive staining (*p* = 0.001, *p* = 0.0498, respectively) (Figure 4a,b,d,e). Neutrophil infiltration (Ly-6B.2), as quantified by the number of cells per field, showed no significant change (Figure 4c,f).

### 2.5. FMT-AD Did Not Increase Serum Amyloid A (SAA) Levels Nor Did It Induce Gut Alterations

Previously, we have demonstrated that TBI can induce alterations in the bloodstream and peripheral organs [51]. Therefore, we analyzed intestinal inflammation to further clarify whether neuropathological changes after TBI are related to the gut environment. SAA is a pro-inflammatory acute-phase protein that increases in the bloodstream after TBI [53,54]. SAA can be produced by the gut and travel through blood vessels to other inflamed tissues, including the brain. We measured the SAA levels in serum at 3 days post-injury in the mice that received FMT-young or FMT-AD. Quantitative western blot analysis showed an increase of 2.5-fold in males after FMT-AD compared to FMT-young, but no increase in females. However, no significant differences were observed (Figure 5a,b). As expected, SAA levels were increased in injured mice compared to sham mice (** *p* < 0.01). To evaluate whether FMT was tolerated in vivo, intestines were collected, washed, fixed, sectioned, and stained with hematoxylin and eosin and Alcian blue to evaluate tissue damage. When comparing the intestines from FMT-young and FMT-AD mice, no differences were observed in either the Alcian blue area or the goblet cells/villus ratio in the small intestine in both male and female mice (Figure 5c,d).

## 3. Discussion

In this study, we demonstrate the significant, and damaging, impact of transplanting fecal microbiota collected from aged AD mice in young control mice following TBI. To demonstrate the detrimental role of the microbiota from AD mice in response to trauma, we evaluated the impact of FMT-AD compared to FMT-young in the recovery of the injured brain of young control mice. We found that FMT-AD worsened lesion size and inflammatory response and that motor behavior deteriorated further after TBI. Despite this, we did not find a significant effect on intestinal damage or inflammation in the periphery in FMT-AD compared to FMT-young mice.

The results from our study indicate that fecal microbiota composition is a key factor for the outcome after brain trauma, adding to the accumulating evidence from human and animal studies which support the hypothesis that the microbiota plays an important role in brain function. Our previous work demonstrated a decrease in the abundance of *Lactobacillus gasseri*, *Ruminococcus flavefaciens*, and *Eubacterium ventriosum* and an increase in the abundance of *Eubacterium sulci* and *Marvinbryantia formatexigens* at 24 h after TBI in mice [55]. Additionally, in animal models of stroke and spinal cord injury, the composition of bacterial taxa has been shown to change in response to central nervous system injury [56,57], and those changes can persist for at least four weeks [56].

The growing awareness of the influential role of the gut microbiome has led to the development of novel treatment strategies based on its modulation. Notably, FMT as a therapeutic procedure involves the transfer of the fecal matter of a healthy individual into a dysbiotic gut in order to restore intestinal flora. FMT has recently been established as an exciting treatment for a variety of disorders and holds great promise to restore neurological function [58,59]. The therapeutic potential of FMT treatment has been shown in an AD mouse model to alleviate pathogenesis [60]. Similarly, FMT rescued the changes in gut microbiota induced by TBI and reverted TBI-induced neurological deficits in rats [61]. FMT from healthy to AD transgenic mice decreased Aβ accumulation and phosphorylation of tau protein via regulating intestinal and systemic immune responses [62]. Conversely, receiving FMT from dysbiotic post-stroke mice has been shown to result in aggravated brain inflammation, lesion volume, and functional deficits [63], suggesting that the gut microbiome can exert its influence via modulation of the brain neuroinflammatory response. Our results suggest that not only can the gut microbiome be modified by a stool transplant after TBI, but that it may be a good target for treatments related to AD, TBI, or both. Given the detrimental link between brain injury and AD, FMT could be a strategy to improve recovery from trauma and reduce peripheral or central inflammation that could, if left untreated, accelerate the pathology of AD.

Another promising treatment strategy used to modulate the microbiota is the administration of probiotics. Multiple studies have illustrated improved memory and behavioral deficits in mice after they received probiotics [61,64]. For example, administration of *Bifidobacterium bifidum* and *Lactobacillus Plantarum* reduced Aβ accumulation, leading to enhanced cognitive function in an AD rat model [65]. Our current results show a trend wherein the levels of *Lactobacillus johnsonii* are decreased in FMT-AD animals compared to their FMT-young counterparts (Figure 2c). These results are further supported by a recent study that shows probiotics increased various *Lactobacillus* species, including *Lactobacillus johnsonii*, which prevented intestinal permeability, modulated proinflammatory factors, and correlated with better behavioral results [66]. Other studies have shown that probiotics may increase the gut microbiome diversity and increase the abundance of Bacteroides and Faecalibacterium [64]. It was also found than probiotics improved memory deficits and reduced cerebral neuronal and synaptic injuries and glial activation in a mouse model of memory deficits and cognitive impairment [67]. Similarly, our current results show that the relative abundance of the genus *Muribaculum* significantly increases in FMT-AD compared to FMT-young mice (Figure 2). Previous studies have observed that this bacterial genus decreases in response to a Western diet [68], and a decrease in *Muribaculum* and an increase in *Lactobacillus* have been shown to occur in a rodent model of Crohn’s disease [69]. Moreover, a recent study has found decreased levels of *Eubacterium rectale*, which is associated with inflammatory processes, in the stool of AD patients [70]. Interestingly, work from our group revealed equivalent results in a cohort of athletes of contact sports following a concussion [71].

The gut microbiome plays an important role in neurological disorders, especially those related to cognitive problems, anxiety, or depression [32]. We also know that intestinal microbial dysbiosis can lead to impaired inflammatory signaling, which has been suggested as a risk factor for brain recovery. A study using a mouse model of cerebral ischemia demonstrated that stroke causes intestinal dysbiosis and a proinflammatory response. Specifically, FMT of a healthy gut microbiome was able to reduce neuroinflammation and other functional deficits caused by stroke [72]. Our work confirms the participation of the brain–gut–microbiome axis in inflammation, as we found a significant increase in glial activation (Figure 4). Specifically, there was greater activation and quantity of microglia, macrophages, and astroglia after FMT-AD in young mice. Another study on inflammation and the gut microbiome found that eradication and limitation of the complexity of the gut microbiota changed the properties of the microglia, specifically their maturation, differentiation, and function [73].

The impact of the gut microbiome can extend past the neuropathology and affect behavioral outcomes. As an example, a recent study showed that FMT was able to rescue behavioral changes after TBI [66]. Intestinal dysbiosis also affected the recovery of motor function after spinal cord injury [56]. Similarly, our study demonstrated that the microbiota of AD mice transplanted into young mice aggravates the motor deficits that originate after TBI (Figure 3d). In addition, the gut microbiome can modulate cognitive behavior [74]. In AD, extracellular Aβ and tau deposits are found in the frontal cortex and the hippocampus [8]. The prefrontal cortex (PFC) has been determined to play a crucial role in learning and response to fear [75,76]. Alterations in the PFC determine cognitive changes in advanced aging, which are indicative of patterns of cognitive dysfunctions observed in patients with AD [77]. Recently, a study linked the gut microbiota transplantation from an AD mouse model into control mice with impaired memory function [78].

Limitations and future directions: We acknowledge that this study has limitations concerning the extent of the colonization of the transplanted bacteria in the host animals and how this is would alter the effect of the FMT after brain trauma. Still, we believe that these questions require long-term studies that are beyond those performed days after TBI. An additional limitation in our study is that we did not have access to aged 3×Tg-AD male mice and all donor animals used for the preparation of the FMT-AD were females, which could have induced a sex-dependent effect. Moreover, young WT animals were used as donors for the FMT-young instead of young 3×Tg-AD mice. Although these could be considered more appropriate controls, we justify the use of young WT because even pre-symptomatic AD mice may have developed a neuroinflammatory response, brain pathology, or gut dysbiosis that would produce confounding effects in our study. Despite having found that FMT-AD increased the neuroinflammation and motor impairment induced by TBI, we did not find any significant change in peripheral inflammation. However, we could observe a trend towards increase that did not reach statistical significance, and an insufficient number of animals may be the reason for not reaching enough statistical power in this assay. It is also possible that the three-day post-injury timepoint was too late, and that the differences in peripheral inflammation could only be detected during the first hours after injury.

Despite the aforementioned limitations, our findings indicate that it is possible to reproduce the effects of the microbiota derived from AD mice by transferring the fecal microbiota into young mice. In future studies with more translational focus, microbiota from human AD patients could be transplanted to mice to characterize their effect following brain trauma. Although the neuropathological response to TBI would continue to exist in mice, characterizing specific bacteria and bacterial relationships identified in both animal and human clinical studies may lead to better interpretation of results. Additionally, our findings suggest that AD gut microbiota alterations may contribute to the mechanisms underlying brain recovery. In this sense, the regulation of brain inflammation through the transplantation of the microbiota of healthy and young mice may be a therapeutic strategy for TBI associated to AD. Consequently, we suggest that FMT from healthy donors to AD patients suffering from TBI could reduce TBI neuroinflammation by remodeling the gut microbiota composition. Future experiments in our laboratory are oriented in this direction.

## 4. Materials and Methods

### 4.1. Mice and Traumatic Brain Injury Model

Young adult (9–12 weeks-old) male and female C57BL/6 mice were purchased from Jackson Laboratories (Bar Harbor, ME, USA). The 3×Tg-AD mice, 18–24 months old, were housed at the Houston Methodist Research Institute animal facilities. We performed qPCR-based genotyping from tail biopsies to confirm hemizygous and WT genotypes for the 3×Tg-AD strain using the Transnetyx, Inc. (Cordova, TN, USA) genotyping service (data not shown). We used an electromagnetically CCI injury device (Impact One impactor CCI, Leica) to induce TBI in the cerebral cortex of C57BL/6 mice. As previously described by our laboratory [50], we applied the following impact coordinates: 2 mm lateral and 2 mm posterior to Bregma with an impact depth of 1.5 mm, using a 2-mm diameter flat impact tip (speed 3.6 m/s, dwell time 100 ms). Sham mice underwent all procedures except impact. Mice were sacrificed 3 days post-injury. The Institutional Animal Care and Use Committee (IACUC) approved all animal studies and breeding protocols at Houston Methodist Research Institute. Studies were conducted following the NRC guide to the Care and Use of Laboratory Animals.

### 4.2. Fecal Microbiota Transplantation (FMT), Fecal Sample Collection, and DNA Extraction

Mice were group-housed based on their experimental group, and daily fecal samples from cages were pooled and redistributed amongst all experimental cages, thereby ensuring microbiome conformity through all mice. Fresh stool pellets were collected from aged 3×Tg-AD donor female mice for FMT. Next, the feces were homogenized in sterile phosphate buffered saline (PBS) and centrifuged at 800× *g* for 3 min to pellet the large particles. The supernatants were collected in fresh sterile tubes, and the final bacterial suspension was stored at −80 °C. A single 200 μL bolus of the prepared microbiota suspension was administered by oral gavage to each recipient C57BL/6 mice (wild-type, WT) at 24 h after TBI. Mice from the control group received a FMT from young WT mice. We used five mice/group with equal numbers of males and females to study sex differences. FMT recipient mice were sacrificed 3 days post-surgery, the abdomen cleaned with 70% ethanol, and the peritoneal cavity opened to expose the cecum. The cecal content was collected in sterile tubes for microbial DNA isolation and metagenomics analysis and stored at −80 °C. Genetic material was isolated from frozen cecum samples using the QIAamp PowerFecal Pro DNA Kit (Qiagen, Germantown, MD, USA). Bead beating was performed for 3 min at 6.5 m/s on a FastPrep-24 system (MP Biomedicals, Irvine, CA, USA). DNA isolation continued as specified in the kit’s instructions.

### 4.3. Long Read 16S rRNA Gene Sequencing

16S rRNA amplicon sequencing was performed on a MinION nanopore sequencer (Oxford Nanopore Technologies, Oxford, UK). The amplicon library was prepared using the 16S Barcoding Kit 1-24 (SQK-16S024, Oxford Nanopore Technologies, Oxford, UK). For the PCR amplification and barcoding, 15 ng of template DNA extracted from fecal samples were added to the LongAmp Hot Start Taq 2X Master Mix (New England Biolabs, Ipswich, MA, USA). Initial denaturation at 95 °C was followed by 35 cycles of 20 s at 95 °C, 30 s at 55 °C, 2 min at 65 °C, and a final extension step of 5 min at 65 °C. Purification of the barcoded amplicons was performed using the AMPure XP Beads (Beckman Coulter, Brea, CA, USA) as per Nanopore’s instructions. Samples were then quantified using a Qubit fluorometer (Life Technologies, Carlsbad, CA, USA) and pooled in an equimolar ratio to a total of 50–100 ng in 10 μL. The pooled library was then loaded into a R9.4.1 flow cell and run per the manufacturer’s instructions. MINKNOW software 19.12.5 was used for data acquisition.

### 4.4. Long-Read 16S Bioinformatic Analysis

Raw sequences were base called with Guppy v4.4.2 with the trim barcodes’ function on and quality score (qscore) filtering set to a minimum score of 7. Microbial community profiles for each sample were generated with Emu v1.0.2 [79], utilizing the default parameters and database. Alpha and beta diversity analyses for microbiome samples were performed with Python3.8 using the sci-kit bio v0.4.1 diversity package. Alpha diversity values were generated for each sample using the Shannon and Simpson indices, while beta diversity was calculated with the Weighted UniFrac metric. The phylogenetic tree was generated from the Emu default database using the taxdump_to_tree.py python script in biocore, with each branch length set to 1. A principal coordinate analysis (PCoA) distance matrix was then generated, and statistical difference between the two groups was calculated with 9999 simulated ANOSIM permutations. To visualize these values, a Matplotlib v.3.3.3 beta diversity scatter plot was generated with 95% confidence ellipses drawn around each group, courtesy of Matplotlib two-dimensional confidence ellipse source code. Stacked bar plots were created with GraphPad Prism version 9 to visualize the average relative abundance of bacteria amongst samples in each group for the given workflow. Log2 fold change was calculated between two samples from the same participant by taking the log2 of the percent relative abundance of the later time point divided by an earlier time point. To avoid mathematical errors, relative abundance values of 0 were transformed to 10^−10^.

### 4.5. Rotarod

Motor performance was assessed by the ability of mice to stay on a rotating rod apparatus (Ugo Basile Harvard Apparatus, PA, USA) as the speed gradually increased. Mice were tested 2 days before injury (baseline) and 1 day post-injury. The rod was accelerated from 4 to 40 rpm in 2 min, and the duration that mice were able to stay on the rod was recorded as latency to fall in seconds.

### 4.6. Cresyl Violet Staining and Lesion Measurements

Brains were fixed in 4% paraformaldehyde overnight and then stored in a 30% sucrose solution at 4 °C. Next, brains were sectioned at 20 µm thickness in coronal orientation through the dorsal hippocampus. The brain sections were cryoprotected in an antifreeze solution (30% glycerol, 30% ethylene glycol in 0.01 M PBS) for storage at −20 °C. Cresyl-violet (0.1%, Sigma-Aldrich, St. Louis, MO, USA) was dissolved in distilled water and filtered. Brain sections were mounted on gelatin-coated glass slides (SuperFrost Plus, Thermo Fisher Scientific, IL, USA) and stained for 20 min with a cresyl-violet solution. Sections were then dehydrated for 2 min sequentially with 100, 95, 70, and then 50% ethanol, cleared in xylene for another 2 min, covered with Permount mounting media (Thermo Fisher Scientific, Waltham, MA, USA), and coverslipped. The lesion area was assessed on 8 to 12 brain sections spaced equidistance (every 450 mm) apart, approximately −1.70 to −2.70 mm from Bregma. Lesion volume was obtained by multiplying the sum of the lesion areas by the distance between sections. The percent of lesion volume was calculated by dividing each lesion volume by the total ipsilateral hemisphere volume (similarly obtained by multiplying the sum of the areas of the ipsilateral hemispheres by the distance between sections). Lesion scores ranged from 0 to 4 (0, no lesion; 1, small cortical lesion; 2, medium cortical lesion; 3, large cortical lesion and small hippocampal lesion; 4, large cortical and hippocampal lesion).

### 4.7. Immunofluorescence Analysis

Free-floating parallel brain sections were washed with PBS + 0.5% Triton X-100 (PBS-T) and then blocked with 5% normal goat serum (NGS, #S1000, Vector laboratories, Burlingame, CA, USA) in PBS-T for 1 h. Brain sections were incubated at 4 °C overnight in PBS-T and 3% of NGS using the following primary antibodies: anti-rabbit Iba-1 (1:1000, Wako) for activated microglia/macrophages, anti-mouse GFAP for astrocytes (1:1000, Millipore, Burlington, MA, USA), or anti-rat Ly-6B.2 alloantigen (1:100, MCA771GA, Bio-Rad Laboratories, Hercules, CA, USA) for neutrophils. After incubation, brain sections were washed in PBS-T and incubated with the corresponding anti-rabbit or anti-mouse Alexa Fluor 568-conjugated and anti-rat Alexa Fluor 488-conjugated IgG secondary antibody (all 1:1000, Thermo Fisher Scientific, Waltham, MA, USA) for 2 h at room temperature. Sections were washed with PBS and incubated with DAPI solution in PBS (1:50,000, Sigma-Aldrich, St. Louis, MO, USA) for counterstained nuclei. The sections were rinsed with distilled water and coverslipped with Fluoro-Gel with Tris Buffer mounting medium (Electron Microscopy Sciences, Hatfield, PA, USA). Images were acquired on a Nikon motorized fluorescence microscope (Eclipse Ni-U, Melville, NY, USA) with a pco. Edge Scomos camera (4.2LT USB3) and analyzed using NIS-Elements software. For quantitative analysis of immunolabeled sections, we implemented unbiased, standardized sampling techniques to measure tissue areas corresponding to the injured cortex showing positive immunoreactivity, as we previously described [50,52]. To quantify the number of Ly6B.2-positive cells in the injured cortex, an average of four coronal sections from the lesion epicenter (−1.34 to −2.30 mm from Bregma) were counted and imaged for each animal, *n* = 5 mice per group. Within each brain region, every positive cell was counted in each of five cortical fields (×20, 151.894 mm^2^) around the impact area, as we have previously described [80].

### 4.8. Western Blot Analysis

For serum collection, trunk blood was collected and allowed to clot for 30 min at room temperature. Next, serum was obtained by centrifugation (1500× *g*) at 4 °C for 20 min and stored at −80 °C. A 1:5 dilution in volume from serum samples was combined with 4 °C Laemmli sample buffer (Bio-Rad Laboratories, Hercules, CA, USA) and heated at 100 °C for 10 min before loading into 12% Mini-PROTEAN TGX Stain-Free gels (Bio-Rad Laboratories, Hercules, CA, USA). Separated proteins were transferred onto nitrocellulose membranes (Bio-Rad Laboratories, Hercules, CA, USA). Following transfer, the membranes were blocked in 5% *w*/*v* skim milk powder in PBS-Tween 20 (PBS-Tw) blocking buffer for 1 h at room temperature. Membranes were incubated with goat anti-SAA primary antibody (1:500; AF2948, R&D Systems, Minneapolis, MN, USA) overnight at 4 °C and with horseradish peroxidase-conjugated rabbit anti-goat secondary antibody (1:3000; Thermo Fisher Scientific, Waltham, MA, USA) for 1 h at room temperature. Membranes were developed with Clarity Western ECL (Bio-Rad Laboratories, Hercules, CA, USA). A ChemiDoc MP imaging system (Bio-Rad Laboratories, Hercules, CA, USA) was used for the stain-free gel. The chemiluminescence imaging and densitometry quantification was performed using ImageLab 6.0.1 software (Bio-Rad Laboratories, Hercules, CA, USA).

### 4.9. Gut Histological Analysis

Intestines from FMT-recipient mice were fixed in 4% paraformaldehyde for 48 h and transferred to 70% ethanol. Tissues were processed in a Shandon Exelsion ES Tissue Processor and embedded in paraffin on a Shandon HistoCenter Embedding System, using the manufacturer’s standard processing and embedding protocols. Slides were sectioned at 5 μm thickness. Paraffin sections were deparaffinized, rehydrated, and then stained with hematoxylin solution for 6 h at a temperature of 60–70 °C. They were then rinsed in tap water until the water was colorless. Next, 0.3% acid alcohol in water was used to differentiate the tissue two times for 2 min, and the sections were rinsed with tap water and stained with eosin for 2 min. Alcian blue stains the mucins produced by the goblet cells of the intestine. For Alcian blue staining, deparaffinized and rehydrated sections were incubated in Alcian blue solution for 30 min and counterstained with 0.1% Nuclear Fast Red for 5 min. Sections were then dehydrated and mounted with permanent mounting media.

### 4.10. Statistical Analysis

To analyze whether individual species or genera exhibited a significant difference in relative abundance between the FMT-AD and FMT-young group, we constructed linear models in MaAsLin2 (MaAsLin2 R package version 1.4.0, http://huttenhower.sph.harvard.edu/maaslin2, accessed on 23 June 2021). Relative abundance from Emu results were used as input, and default transformation and outcome distribution options were selected. No normalization methods were utilized since input data were already in relative abundance form. Models were run separately for species- and genus-level analysis. For both models, FMT donor was the fixed effect (*n* = 9 FMT-young, 8 FMT-AD), and a threshold for a relative abundance of at least 1% in a minimum of 10% of the samples was applied in order to reduce hypothesis tests requiring correction. For behavioral and immunostaining data analysis, we used two-way or three-way analysis of variance (ANOVA) followed by a Sidak posthoc test. An unpaired *t*-test was used to determine statistical differences in the gut histology quantifications. Data were represented as the mean and standard error of the mean (±SEM), and *p*-values < 0.05 were considered statistically significant. GraphPad Prism 9 software (GraphPad Software, San Diego, CA, USA) was used for statistical analysis. All mice were randomized to experimental conditions, and experimenters were blind to genotype and treatment groups throughout the study.

## 5. Conclusions

Our study provides novel insight into the microbiota function in the neuropathology following brain injury. We demonstrated how the gut microbiota from AD mice aggravates the TBI outcomes, supporting the idea that the diversity and composition of the gut microbiome affects the impact of and recovery from neuroinflammatory disorders, specifically brain trauma in a preclinical mouse model. It provides support for the hypothesis that microbe-based approaches that aim to restore youth-like microbiota could improve cognitive function and improve recovery after brain damage. In the future, it will be interesting to see if restoration of the gut microbiota of AD patients is also able to reduce the progression of brain damage. This represents a shift in AD research, from the amyloid hypothesis to the microbial features underlying the disease. Understanding the microbiota effects in the context of the recovery of people with AD or AD-associated dementia (an increasingly prevalent demographic segment of modern societies who also suffer head trauma) may have significant future implications for advancing their quality of life. Future research on brain–gut axis involvement in AD-associated TBI is necessary for new treatment targets and therapies.

## Figures and Tables

**Figure 1 ijms-23-04476-f001:**
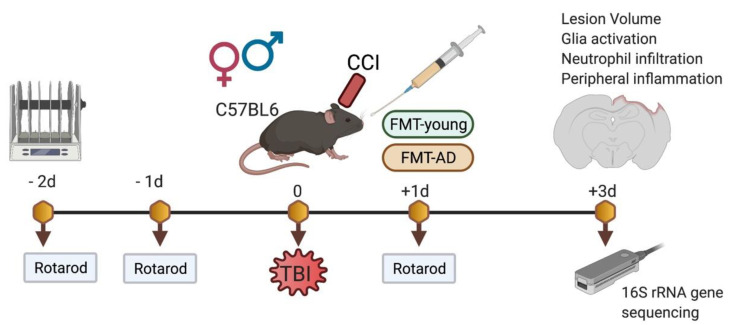
Experimental design. The controlled cortical injury (CCI, a model of TBI) was performed on WT mice. FMT-young and FMT-AD groups received fecal matter transplants by oral gavage from young WT and aged 3×Tg-AD mice, respectively. Motor performance on the rotarod test was assessed before the injury and at 1 day post-TBI. Brains, blood, and colon samples were collected 3 days post-TBI for immunohistochemical and molecular characterization. Gut microbiome composition from cecal contents was analyzed by 16S rRNA gene sequencing using the Nanopore platform.

**Figure 2 ijms-23-04476-f002:**
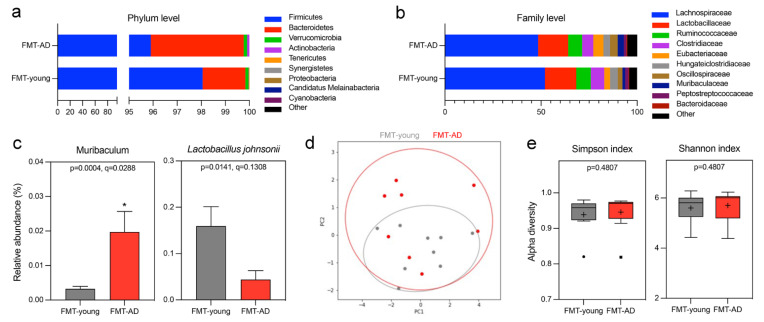
Gut microbiome changes after FMT from aged AD mice. (**a**,**b**) Relative abundances of the top 9 phyla (**a**) and families (**b**) in the cecum microbiota of mice that received FMT from young WT (FMT-young) or aged AD (FMT-AD) mice. (**c**) Relative abundances of the genera and species show significant alterations in the random-effects multivariate analysis for FMT-young (*n* = 9, gray) and FMT-AD (*n* = 8, red). An abundance cutoff of >0.1% was applied at the species level. * *q* < 0.05. (**d**) Principal Coordinate Analysis (PCoA) ordination plot based on Weighted UniFrac distances for FMT-young (*n* = 9, gray) and FMT-AD (*n* = 8, red). The confidence ellipses represent the 95% confidence interval. Analysis of Similarity (ANOSIM) evaluated group dissimilarities. (**e**) Shannon and Simpson alpha diversity indices at the species level for FMT-young (*n* = 9, gray) and FMT-AD (*n* = 8, red). In the box and whisker plots, the cross represents the mean. Significance was determined by Kruskal–Wallis, followed by Dunn’s multiple comparisons test.

**Figure 3 ijms-23-04476-f003:**
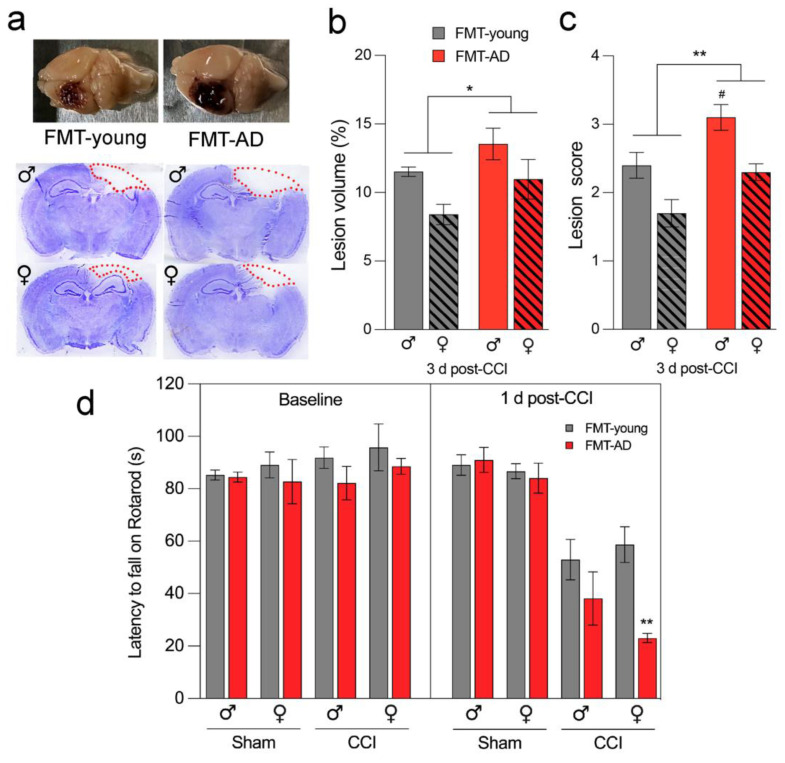
Microbiota from AD mice aggravates lesion size and impairs motor ability after TBI. (**a**) Representative images (**top**) and sections stained with cresyl-violet (**bottom**) of injured brains at 3 days post-injury. The dotted line indicates the lesion area composed of the cavity and edematous area. (**b**,**c**) FMT from AD mice significantly increased the mean lesion volume compared with lesion volume in mice that received FMT from young animals. *n* = 5. * *p* < 0.05, ** *p* < 0.01 (ANOVA); ^#^
*p* < 0.05 (post-hoc test). (**d**) Rotarod motor test shows that FMT from AD mice further impaired motor ability 1 day post-TBI compared to FMT from young mice. At the 1 d post-CCI timepoint, three-way ANOVA shows significance for the sham/CCI and FMT-young/FMT-AD comparisons (*p* < 0.0001 and *p* = 0.0037, respectively) but not for sex (*p* = 0.2579). *n* = 10, ** *p* < 0.01 (post-hoc).

**Figure 4 ijms-23-04476-f004:**
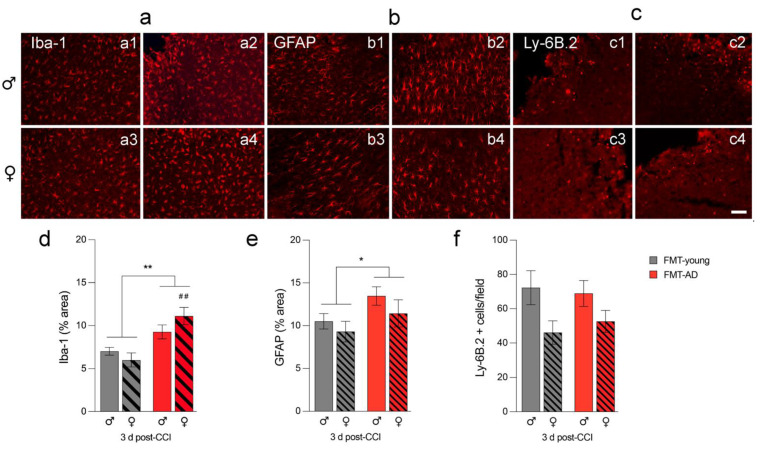
Microbiota from AD mice increased neuroinflammation in injured C57BL/6 mice. Glia activation increases in the pericontusional cortex after fecal matter transplants from AD mice (FMT-AD) compared to FMT from young mice (FMT-young). Representative immunofluorescence images indicate that FMT-AD compared to FMT-young induces an increased inflammatory response in microglia/macrophages (Iba-1; **a**(**a1**–**a4**)) and astroglia (GFAP; **b**(**b1**–**b4**)), but no change in neutrophils (Ly-6B.2; **c**(**c1**–**c4**)) at 3 days post-injury. The scale bar is 50 μm. The quantification graphs show a significantly higher area occupied by Iba-1 (**d**) and GFAP (**e**) positive cells in FMT recipients from AD mice. (**f**) The number of cells Ly-6B.2 not significantly changed in the FMT-AD mice. *n* = 5. * *p* < 0.05, ** *p* < 0.01 (ANOVA), ^##^
*p* < 0.01 (post-hoc test).

**Figure 5 ijms-23-04476-f005:**
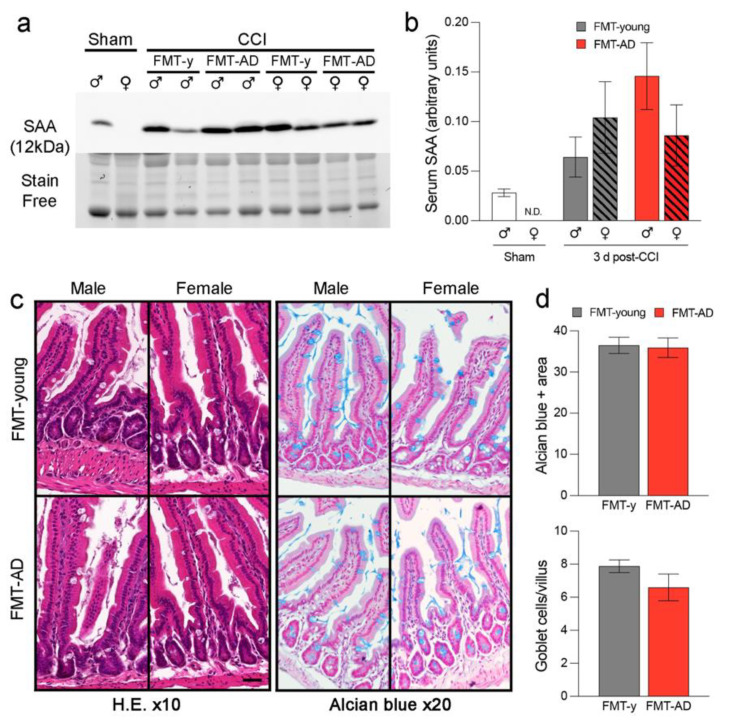
FMT-AD neither induced peripheral inflammation nor caused intestinal damage to recipient mice. (**a**) Western blot showing protein Serum Amyloid A (SAA) levels in the serum from FMT-AD and FMT-young (FMT-y) mice at 3 d post-TBI. (**b**) Quantification of the western blot shows a not statistically significant increase in SAA in the serum of males after FMT-AD compared to FMT-y. (**c**) Representative histology images from intestinal tissue do not show any abnormality between the FMT-y and FMT-AD groups in male or female mice. Left, Hematoxylin and eosin (H.E.) staining; right (×10 Objective), Alcian blue staining (×20 Objective). (**d**) Densiometric quantification of the Alcian blue positive area and counting the number of goblet cells per villus are not different between FMT-y and FMT-AD groups. Male and female data were combined, since two-way ANOVA showed no sex differences (Alcian blue area, *p* = 0.429; goblet cells, *p* = 0.471).

## Data Availability

The 16S rRNA sequencing data has been deposited in SRA under BioProject PRJNA826428.

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
