# Peer review of "Fecal Microbiota Transplantation Derived from Alzheimer’s Disease Mice Worsens Brain Trauma Outcomes in Wild-Type Controls"

_ijms, 2022, doi:10.3390/ijms23094476_

Round 1

Reviewer 1 Report

Soriano S. et al. have investigated the effect of fecal microbiota transplants (FMT) from AD mice in young WT mice after traumatic brain injury (TBI). This is a well-written and interesting paper. I have only few comments/suggestions:

  • Figure 3: In the Rotarod motor test, male and female data were combined. Since FMT from AD mice significantly increased the mean lesion volume more in males than in females 3 days post-injury (figure 3b and c), authors should provide this data for both sexes (not pooled).
  • Figure 4: Authors should indicate which was the brain region that they have imaged for GFAP, Iba1 and Ly-6B.2 stainings. Why the authors decided to quantify the % area for GFAP and Iba1 and instead, number of cell/field for Ly-6B.2? Moreover, in each image dapi is needed.
  • Authors should specify in the material and methods section the sex of 3xTg-AD used for FMT collection.
  • Authors should consider to discuss why they have decided to use FMT from young WT animal as control instead that FMT-AD coming from young animals without AD pathology/plaques.

Author Response

REVIEWER 1:

We would like to thank the reviewer for the constructive comments and valuable suggestions that have helped us improve the manuscript. Below we provide point-by-point responses to the reviewer’s comments and we have made edits to the manuscript accordingly (highlighted in yellow).

Soriano S. et al. have investigated the effect of fecal microbiota transplants (FMT) from AD mice in young WT mice after traumatic brain injury (TBI). This is a well-written and interesting paper. I have only few comments/suggestions:

  • Figure 3: In the Rotarod motor test, male and female data were combined. Since FMT from AD mice significantly increased the mean lesion volume more in males than in females 3 days post-injury (figure 3b and c), authors should provide this data for both sexes (not pooled).

Response: We thank the reviewer for the suggestion, the motor behavioral testing (Rotarod) results have been split by sex and included in Figure 3d. Sex differences have been incorporated into the results section.

  • Figure 4: Authors should indicate which was the brain region that they have imaged for GFAP, Iba1, and Ly-6B.2 stainings. Why the authors decided to quantify the % area for GFAP and Iba1 and instead, number of cell/field for Ly-6B.2? Moreover, in each image dapi is needed.

Response: Following the reviewer’s indication, we have specified in the results section and in the corresponding figure legend that the brain region imaged was the pericontusional cortex.

In response to brain injury, there is an increase in astrocyte reactivity, which translates into a change in morphology represented by the area occupied by the cells in proportion to the total area (% area). These reactive astrocytes acquire a hypertrophic morphology, involving the extension of processes and swelling of cell bodies. To assess this hyperactivated state, we measure the percentage area for the marker GFAP, which is a synonym of activation and not cellular density. Similarly, the microglia acquire an activated ramified morphology after brain injury that can be quantified as an increased % area for the Iba-1 marker (Villapol et al., 2014, PMID: 24926283; Karve et al., 2016, PMID: 25752446). However, in the case of Ly-6B.2, we are evaluating the infiltration of neutrophils into the cortex induced by brain injury, which translates into an increase in the number of this cell type (Wicker et al., 2019, PMID: 31119176). The total number of cells counted as nuclei was not used for any of the quantifications of GFAP, Iba-1, and Ly-6B.2. Therefore DAPI nuclei contrast does not provide further information as it was not included in the quantitative analyses.

  • Authors should specify in the material and methods section the sex of 3xTg-AD used for FMT collection.

Response: Thanks for pointing this out. 3xTg-AD females were used as donors for the FMT collection.  These aged transgenic mice (18 to 24-month-old) were a gift from a fellow investigator, as we indicated in the acknowledgments section, and they only had females available. We have added this information to the methods and discussion sections.    

  • Authors should consider to discuss why they have decided to use FMT from young WT animal as control instead that FMT-AD coming from young animals without AD pathology/plaques.

Response: We appreciate the thoughtful recommendation by the reviewer. We have decided to use FMT from young WT as a control group because 3xTg-AD mice develop neuropathological features at two months old. Furthermore, it is unknown when the changes in the composition of the gut microbiota occur in 3xTg-AD mice that could affect the effects of FMT as controls. We decided to use wild-type mice as controls to ensure that our control group did not have any inflammatory components affecting AD pathology or intestinal dysbiosis. We have added this justification in the discussion of the manuscript.

Reviewer 2 Report

26 March 2022

Regarding the review of manuscript “Fecal microbiota transplantation derived from Alzheimer's disease mice worsens brain trauma outcomes in young C57BL/6 mice” by Soriano S et al., submitted to International Journal of Molecular Sciences

Manuscript ID: ijms-1656063

Dear Authors,

Soriano and colleagues investigated in the present study entitled ‘Fecal microbiota transplantation derived from Alzheimer's disease mice worsens brain trauma outcomes in young C57BL/6 mice’, the current status of knowledge of the link between gut microbiota from Alzheimer’s disease (AD) and neurological deficits after traumatic brain injury (TBI) in young mice. For this purpose, the authors performed fecal microbiota transplants from AD (FMT-AD) mice into young C57BL/6 (wild-type, WT) mice following TBI, and analyzed the full-length rRNA sequences from mouse fecal samples. The results showed that FMT-AD administration stimulated a higher relative abundance of Muribaculum intestinal and a decrease in Lactobacillus johnsonii; furthermore, WT mice exhibited larger lesions, increased activated microglia/macrophages, and reduced motor recovery after FMT-AD compared to FMT-young one day after TBI. The authors concluded by stating that the gut microbiota from AD mice not only aggravated the neuroinflammatory response and motor recovery but also increased the lesion size after TBI in young WT mice.

The main strength of this original research article is that it addresses an interesting and timely question, investigating how microbiota from AD mice aggravates the TBI outcomes. In general, I think the idea of this article is really interesting and the authors’ fascinating observations on this timely topic may be of interest to the readers of the International Journal of Molecular Sciences. However, some comments, as well as some crucial evidence that should be included to support the author’s argumentation, needed to be addressed to improve the quality of the manuscript, its adequacy, and its readability prior to the publication in the present form, in particular reshaping parts of the Introduction and Discussion sections by adding more evidence and theoretical constructs.

Please consider the following comments:

  1. Abstract:
  2. Please proportionally present background, purpose, methods, results, and conclusion.
  3. I suggest clearly presenting the type of AD model mice, age, and the source of the microbiota.
  4. Also the description of traumatic injury model including cortical impact injury device is necessary.
  5. Keywords: I recommend listing up to ten keywords.
  6. A graphical abstract is highly recommended.
  7. In general, I recommend authors to use more evidence to back their claims, especially in the Introduction of the article, which I believe is currently lacking. Thus, I recommend the authors to attempt to deepen the subject of their manuscript, as the bibliography is too concise: nonetheless, in my opinion, less than 60/70 articles for a research paper are really insufficient. Indeed, currently, authors cite only 53 papers, and they are too low. Therefore, I suggest the authors to focus their efforts on researching more relevant literature: I believe that adding more studies and reviews will help them to provide better and more accurate background to this study. In this review, I will try to help the authors by suggesting some relevant literature of my knowledge that suits their manuscript.
  8. Introduction:
  9. I suggest the authors to reshape the Introduction section, which seems inhomogeneous and dispersive. First of all, I suggest presenting the brief descriptions of the involvement of microbiota in neurologic and psychiatric diseases in general, the mechanism including inflammation, and recent research in animal models (https://doi.org/10.3390/microorganisms9112281; https://doi.org/10.3390/biomedicines10020446; https://doi.org/10.3390/ph14040340; https://doi.org/10.3390/brainsci11081085; https://doi.org/10.3390/cells9112476; https://doi.org/10.3390/brainsci11081038). Information about the pathophysiology of AD is completely missing, specifically its definition, causes, symptoms and related neurocognitive changes. Considering that this study's main focus is to deepen the current understanding of Alzheimer's disease pathogenesis and risk factors, I suggest the authors to make such effort to provide a brief overview of the pertinent published literature that offers a perspective on pathophysiology and neurological changes of AD, because as it stands, there is no mention of this in the manuscript. To this end, to gain a more comprehensive and appropriate theoretical background on this topic, I would recommend citing a review that examined pathophysiological basis and biomarkers of AD pathology (https://doi.org/10.3390/ijms21249338) and a study in which authors investigated age-related impairments in the ability to process contextual information and in the regulation of responses to threat, addressing that structural and physiological alterations in the prefrontal cortex and medial temporal lobe determine cognitive changes in advanced aging, that can eventually cause patterns of cognitive dysfunctions observed in patients with AD/MCI (https://doi.org/10.1038/s41598-018-31000-9). Furthermore, I would recommend adding information on how gut microbiota changes correlate with cognitive (i.e., dysfunction in attention and emotion perception) and neuropsychiatric symptoms in neurologic patients (https://doi.org/10.3390/biomedicines10030627). I firmly believe that this evidence will help to provide a more coherent and defined background.
  10. In according with the previous point raised, when authors stated that ‘AD is characterized by progressive cognitive and motor impairment associated with the accumulation of beta-amyloid (Aβ) protein and tau protein deposition’, I would suggest adding some studies that might discuss amyloid-β (Aβ) pathology in AD, highlighting the combined effect of forms of Aβ and tau protein to drive healthy neurons into the diseased state. Aβ peptide and tau protein consistently accumulate in the frontal and/or parietal lobes and cause alterations of the frontal lobe that impact memory and error-driven learning in individuals who have a high risk of dementia: evidence from a recent theoretical review (https://doi.org/10.1038/s41380-021-01326-4) that focused on the neurobiology of fear conditioning, analyzed the role of the ventromedial prefrontal cortex (vmPFC) was analyzed in the processing of safety-threat information and their relative value, and how this region is fundamental for the evaluation and representation of stimulus-outcome’s value needed to produce sustained physiological responses. Also, I believe that a recent yet relevant perspective manuscript (https://doi.org/10.17219/acem/146756) might be of interest: here the focus was on providing a deeper understanding of human learning neural networks, particularly on human PFC crucial role, that might also contribute to the advancement of alternative, more precise and individualized treatments for psychiatric disorders. Secondary, authors also might to consider some studies that have focused on this topic (https://doi.org/10.3390/biomedicines10010076; https://doi.org/10.3390/biomedicines9050517). I believe that adding information from these studies may improve the theoretical background of the present article and its argumentation by highlighting how cognitive alterations caused by frontal dysfunction are fundamental as neurodegenerative biomarkers of AD.
  11. The objectives are generally clear; however, I believe that there are some ambiguous points that require clarification or refining. I think that authors here need to be explicit regarding how they operationally determined the effect of AD microbiota on recovery following TBI. Furthermore, I think that the authors should be explicit regarding the hypothesis of a direct link between fecal microbiota transplants from AD (FMT-AD) and neuroinflammation.
  12. Results: In my opinion, this section is well organized, but it illustrates findings in an excessively broad way, without really providing full statistical details, to ensure in-depth understanding and replicability of the findings. Specifically, provide more detail about levels of inflammatory necrosis and altered levels of glutathione, because it appears unclear and hard to grasp how to interpret sets of glycoproteins. Also, in my opinion, it is necessary for the authors to present their findings using summary tables.
  13. Discussion: In this final section, the authors described the results and their argumentation and captured the state of the art well; however, I would have liked to see some views on a way forward. I believe that the authors should make an effort, trying to explain the theoretical implication as well as the translational application of this research article, to adequately convey what they believe is the take-home message of their study. Discussion of theoretical and methodological avenues in need of refinement is necessary, as well as suggestions of a path forward in understanding the involvement of brain-gut-microbiome axis in the neuropathology following brain injury, with a focus on how it may be possible to impact the recovery from neuroinflammatory disorders. In this regard, recent evidence suggests that the application of new methods in Neuroinflammatory disorders’ treatment, such as the Non-invasive brain stimulation techniques (NIBS), have shown promising results in humans (https://doi.org/10.3390/brainsci12030364). Importantly, I recommend referring to recent studies that revealed that the application of NIBS induces long-lasting effects, noninvasively modulating the cortical excitability, and modulating a variety of cognitive functions: for example, a recent review acknowledged the implementation of NIBS to modulate in general fear memories (https://doi.org/10.1016/j.neubiorev.2021.04.036). Additionally, I would suggest another recent study that illustrated the therapeutic potential of NIBS as a valid alternative for those patients not responding or drug treatments (https://doi.org/10.1016/j.jad.2021.02.076). In addition to the previously mentioned literature, authors might also see these additional studies that have focused on the efficacy of NIBS and IBS (https://doi.org/10.3389/fpsyt.2018.00201; https://doi.org/10.3389/fnagi.2020.578339).
  14. In my opinion, I think the ‘Conclusions’ paragraph would benefit from some thoughtful as well as in-depth considerations by the authors, because as it stands, it is very descriptive but not enough theoretical as a discussion should be. Authors should make an effort, trying to explain the theoretical implication as well as the translational application of their research.
  15. In according to the previous comment, I would ask the authors to include a ‘Limitations and future directions’ section before the end of the manuscript, in which authors can describe in detail and report all the technical issues brought to the surface.
  16. Figures: I suggest modifying all figures for clarity and provide higher-quality images because, as it stands, the readers may have difficulty comprehending them. In my opinion, data settings are written with a very small font. Also, please change the scale of the vertical axis and use the same minimum/maximum scale value in all the graphs in all the figures and reorganize the graphs’ space, to provide a better understanding and a direct interpretation of the results.
  17. References: According to the Journal’s guidelines, authors should have provided the DOI number for each reference.

Overall, the manuscript contains no table, three figures and 42 references. In my opinion, the number of references it is too low for an original research article, and this issue may prevent the possibility of publishing it in this form. However, I believe that the manuscript may carry important value investigating how gut microbiota from AD mice aggravates the TBI outcomes, supporting the idea that the diversity and composition of the gut microbiome affects the impact of and recovery from neuroinflammatory disorders.

I hope that, after these careful revisions, the manuscript can meet the Journal’s high standards for publication. I am available for a new round of revision of this review.

Best regards,

Reviewer

Author Response

REVIEWER 2: 

Comments and Suggestions for Authors

26 March 2022

Regarding the review of manuscript “Fecal microbiota transplantation derived from Alzheimer's disease mice worsens brain trauma outcomes in young C57BL/6 mice” by Soriano S et al., submitted to International Journal of Molecular Sciences

Manuscript ID: ijms-1656063

Dear Authors,

Soriano and colleagues investigated in the present study entitled ‘Fecal microbiota transplantation derived from Alzheimer's disease mice worsens brain trauma outcomes in young C57BL/6 mice’, the current status of knowledge of the link between gut microbiota from Alzheimer’s disease (AD) and neurological deficits after traumatic brain injury (TBI) in young mice. For this purpose, the authors performed fecal microbiota transplants from AD (FMT-AD) mice into young C57BL/6 (wild-type, WT) mice following TBI, and analyzed the full-length rRNA sequences from mouse fecal samples. The results showed that FMT-AD administration stimulated a higher relative abundance of Muribaculum intestinal and a decrease in Lactobacillus johnsonii; furthermore, WT mice exhibited larger lesions, increased activated microglia/macrophages, and reduced motor recovery after FMT-AD compared to FMT-young one day after TBI. The authors concluded by stating that the gut microbiota from AD mice not only aggravated the neuroinflammatory response and motor recovery but also increased the lesion size after TBI in young WT mice.

The main strength of this original research article is that it addresses an interesting and timely question, investigating how microbiota from AD mice aggravates the TBI outcomes. In general, I think the idea of this article is really interesting and the authors’ fascinating observations on this timely topic may be of interest to the readers of the International Journal of Molecular Sciences. However, some comments, as well as some crucial evidence that should be included to support the author’s argumentation, needed to be addressed to improve the quality of the manuscript, its adequacy, and its readability prior to the publication in the present form, in particular reshaping parts of the Introduction and Discussion sections by adding more evidence and theoretical constructs.

Response: We were pleased that the reviewer found our article interesting, and we are grateful for the time and consideration that the reviewer dedicated to their comments. First, we would like to thank the reviewer for his/her constructive comments, valuable suggestions, scientific input, and the time they have dedicated to helping us improve our manuscript. We have edited the manuscript following all the reviewers' comments and tracked the changes in the manuscript document (highlighted in yellow). After addressing all the points raised by the reviewers, title edits, re-writing text, changing figures, and adding the requested references. We believe our manuscript has been substantially improved and we hope you will find the revised version better fits the expectations of the journal’s readership and thus is now suitable for publication.

Please consider the following comments:

  1. Abstract:
  2. Please proportionally present background, purpose, methods, results, and conclusion.
  3. I suggest clearly presenting the type of AD model mice, age, and the source of the microbiota.

Also the description of traumatic injury model including cortical impact injury device is necessary. Response: We thank the reviewer for the recommendation, we have edited the abstract accordingly.

  1. Keywords: I recommend listing up to ten keywords.

Response: Following the reviewer suggestion, we have added 10 keywords.

  1. graphical abstract is highly recommended.

Response: An updated version of the graphical abstract has been submitted.

  1. In general, I recommend authors to use more evidence to back their claims, especially in the Introduction of the article, which I believe is currently lacking. Thus, I recommend the authors to attempt to deepen the subject of their manuscript, as the bibliography is too concise: nonetheless, in my opinion, less than 60/70 articles for a research paper are really insufficient. Indeed, currently, authors cite only 53 papers, and they are too low. Therefore, I suggest the authors to focus their efforts on researching more relevant literature: I believe that adding more studies and reviews will help them to provide better and more accurate background to this study. In this review, I will try to help the authors by suggesting some relevant literature of my knowledge that suits their manuscript.

Response: We appreciate the reviewer sharing relevant articles as this has allowed us to substantially improve our manuscript. We have now added 80 references and rewritten important parts of the introduction. 

  1. Introduction:
  2. I suggest the authors to reshape the Introduction section, which seems inhomogeneous and dispersive. First of all, I suggest presenting the brief descriptions of the involvement of microbiota in neurologic and psychiatric diseases in general, the mechanism including inflammation, and recent research in animal models (https://doi.org/10.3390/microorganisms9112281; https://doi.org/10.3390/biomedicines10020446; https://doi.org/10.3390/ph14040340; https://doi.org/10.3390/brainsci11081085; https://doi.org/10.3390/cells9112476; https://doi.org/10.3390/brainsci11081038). Information about the pathophysiology of AD is completely missing, specifically its definition, causes, symptoms and related neurocognitive changes. Considering that this study's main focus is to deepen the current understanding of Alzheimer's disease pathogenesis and risk factors, I suggest the authors to make such effort to provide a brief overview of the pertinent published literature that offers a perspective on pathophysiology and neurological changes of AD, because as it stands, there is no mention of this in the manuscript. To this end, to gain a more comprehensive and appropriate theoretical background on this topic, I would recommend citing a review that examined pathophysiological basis and biomarkers of AD pathology (https://doi.org/10.3390/ijms21249338) and a study in which authors investigated age-related impairments in the ability to process contextual information and in the regulation of responses to threat, addressing that structural and physiological alterations in the prefrontal cortex and medial temporal lobe determine cognitive changes in advanced aging, that can eventually cause patterns of cognitive dysfunctions observed in patients with AD/MCI (https://doi.org/10.1038/s41598-018-31000-9). Furthermore, I would recommend adding information on how gut microbiota changes correlate with cognitive (i.e., dysfunction in attention and emotion perception) and neuropsychiatric symptoms in neurologic patients (https://doi.org/10.3390/biomedicines10030627). I firmly believe that this evidence will help to provide a more coherent and defined background.

Response: We greatly appreciate these thoughtful recommendations. Major edits have been made to the introduction to improve the flow and incorporate the topics and the citations suggested by the reviewer here and in later points. 

  1. In according with the previous point raised, when authors stated that ‘AD is characterized by progressive cognitive and motor impairment associated with the accumulation of beta-amyloid (Aβ) protein and tau protein deposition’, I would suggest adding some studies that might discuss amyloid-β (Aβ) pathology in AD, highlighting the combined effect of forms of Aβ and tau protein to drive healthy neurons into the diseased state. Aβ peptide and tau protein consistently accumulate in the frontal and/or parietal lobes and cause alterations of the frontal lobe that impact memory and error-driven learning in individuals who have a high risk of dementia: evidence from a recent theoretical review (https://doi.org/10.1038/s41380-021-01326-4) that focused on the neurobiology of fear conditioning, analyzed the role of the ventromedial prefrontal cortex (vmPFC) was analyzed in the processing of safety-threat information and their relative value, and how this region is fundamental for the evaluation and representation of stimulus-outcome’s value needed to produce sustained physiological responses. Also, I believe that a recent yet relevant perspective manuscript (https://doi.org/10.17219/acem/146756) might be of interest: here the focus was on providing a deeper understanding of human learning neural networks, particularly on human PFC crucial role, that might also contribute to the advancement of alternative, more precise and individualized treatments for psychiatric disorders. Secondary, authors also might to consider some studies that have focused on this topic (https://doi.org/10.3390/biomedicines10010076; https://doi.org/10.3390/biomedicines9050517). I believe that adding information from these studies may improve the theoretical background of the present article and its argumentation by highlighting how cognitive alterations caused by frontal dysfunction are fundamental as neurodegenerative biomarkers of AD.

Response: We thank the reviewer for this suggestion, we have incorporated this perspective to the discussion since we considered it was a better fit, and we have also added the references suggested.

  1. The objectives are generally clear; however, I believe that there are some ambiguous points that require clarification or refining. I think that authors here need to be explicit regarding how they operationally determined the effect of AD microbiota on recovery following TBI. Furthermore, I think that the authors should be explicit regarding the hypothesis of a direct link between fecal microbiota transplants from AD (FMT-AD) and neuroinflammation.

Response: Thank you for the recommendation, we have further clarified the objectives in the introduction of the manuscript.

  1. Results: In my opinion, this section is well organized, but it illustrates findings in an excessively broad way, without really providing full statistical details, to ensure in-depth understanding and replicability of the findings. Specifically, provide more detail about levels of inflammatory necrosis and altered levels of glutathione, because it appears unclear and hard to grasp how to interpret sets of glycoproteins. Also, in my opinion, it is necessary for the authors to present their findings using summary tables.

Response: Following the reviewer suggestion, we have checked and updated the statistical details in the results section.

  1. Discussion: In this final section, the authors described the results and their argumentation and captured the state of the art well; however, I would have liked to see some views on a way forward. I believe that the authors should make an effort, trying to explain the theoretical implication as well as the translational application of this research article, to adequately convey what they believe is the take-home message of their study. Discussion of theoretical and methodological avenues in need of refinement is necessary, as well as suggestions of a path forward in understanding the involvement of brain-gut-microbiome axis in the neuropathology following brain injury, with a focus on how it may be possible to impact the recovery from neuroinflammatory disorders. In this regard, recent evidence suggests that the application of new methods in Neuroinflammatory disorders’ treatment, such as the Non-invasive brain stimulation techniques (NIBS), have shown promising results in humans (https://doi.org/10.3390/brainsci12030364). Importantly, I recommend referring to recent studies that revealed that the application of NIBS induces long-lasting effects, noninvasively modulating the cortical excitability, and modulating a variety of cognitive functions: for example, a recent review acknowledged the implementation of NIBS to modulate in general fear memories (https://doi.org/10.1016/j.neubiorev.2021.04.036). Additionally, I would suggest another recent study that illustrated the therapeutic potential of NIBS as a valid alternative for those patients not responding or drug treatments (https://doi.org/10.1016/j.jad.2021.02.076). In addition to the previously mentioned literature, authors might also see these additional studies that have focused on the efficacy of NIBS and IBS (https://doi.org/10.3389/fpsyt.2018.00201; https://doi.org/10.3389/fnagi.2020.578339).

Response: We appreciate the reviewer’s suggestions and we have edited the discussion and added these and other new references.

.

  1. In my opinion, I think the ‘Conclusions’ paragraph would benefit from some thoughtful as well as in-depth considerations by the authors, because as it stands, it is very descriptive but not enough theoretical as a discussion should be. Authors should make an effort, trying to explain the theoretical implication as well as the translational application of their research.

Response: We thank the reviewer for the suggestion. We aimed at keeping the conclusion paragraph concise but still including theoretical implications, such as the shift in the AD paradigm centered on the amyloid hypothesis to including the gut microbiome as key factor in the pathogenic cascade. We also included the translational application of our findings, leading to further research on gut microbiota restoration as therapeutic avenue in AD patients suffering from brain trauma. Further considerations about these aspects were more detailed in the discussion section that precedes the conclusion paragraph. 

  1. In according to the previous comment, I would ask the authors to include a ‘Limitations and future directions’ section before the end of the manuscript, in which authors can describe in detail and report all the technical issues brought to the surface.

Response: Thank you for pointing this out, we have added a limitations and future directions paragraph. 

  1. Figures: I suggest modifying all figures for clarity and provide higher-quality images because, as it stands, the readers may have difficulty comprehending them. In my opinion, data settings are written with a very small font. Also, please change the scale of the vertical axis and use the same minimum/maximum scale value in all the graphs in all the figures and reorganize the graphs’ space, to provide a better understanding and a direct interpretation of the results.

Response: We thank the reviewer for bringing this to our attention. We have increased the overall size of all the figures, which also made the font size bigger and easier to read. The vertical axis was changed to use the same min/max scale for the graphs where the unit was the same.

  1. References: According to the Journal’s guidelines, authors should have provided the DOI number for each reference.

Response: Thanks for pointing this out, the references now have the DOI number and are formatted following the Journal’s style.

Overall, the manuscript contains no table, three figures and 42 references. In my opinion, the number of references it is too low for an original research article, and this issue may prevent the possibility of publishing it in this form. However, I believe that the manuscript may carry important value investigating how gut microbiota from AD mice aggravates the TBI outcomes, supporting the idea that the diversity and composition of the gut microbiome affects the impact of and recovery from neuroinflammatory disorders.

I hope that, after these careful revisions, the manuscript can meet the Journal’s high standards for publication. I am available for a new round of revision of this review.

Response: We greatly appreciate the encouraging words from the reviewer regarding our study. We have increased the number of references including the articles suggested by the reviewer and others that we found upon further literature review. We have 80 references now. Also, we have edited the manuscript according to the reviewer’s suggestions and performed major updates in the introduction and discussion sections. We believe the manuscript has been substantially improved and we also hope the revised version better fits the expectations of the IJMS journal and thus is now suitable for publication.

Round 2

Reviewer 2 Report

15 April 2022

Regarding the 2nd review of manuscript “Fecal microbiota transplantation derived from Alzheimer's disease mice worsens brain trauma outcomes in young C57BL/6 mice” by Soriano S et al., submitted to International Journal of Molecular Sciences (IJMS)

Manuscript ID: ijms-1656063

Dear Authors,

I am pleased to see that the authors did an excellent work clarifying the questions I have raised in the previous round of review. Currently, this paper is a well-written, timely piece of research and provides a useful study addressing an interesting and innovative question, investigating how microbiota from a mouse model of Alzheimer’s disease aggravates the outcomes of traumatic brain injury.

The manuscript contains five figures, no table, and 80 references. Overall, this is a timely and needed work. Thus, I believe that manuscript now meets the Journal’s standards for publication. I am always available for other reviews of such interesting and important articles. I look forward to seeing further study on this issue by these authors in the future.

Thank you for your work.

Best regards,

Reviewer